

# Accelerating Lagrangian transport simulations on graphics processing units: performance optimizations of MPTRAC v2.6

Lars Hoffmann[1,2], Kaveh Haghighi Mood[1], Andreas Herten[1], Markus Hrywniak[3], Jiri Kraus[3], Jan Clemens[1,2,4], and Mingzhao Liu[1,2]

[1]Jülich Supercomputing Centre, Forschungszentrum Jülich, Jülich, Germany
[2]Center for Advanced Simulation and Analytics (CASA), Forschungszentrum Jülich, Jülich, Germany
[3]NVIDIA GmbH, Würselen, Germany
[4]Institut für Energie- und Klimaforschung (IEK-7), Forschungszentrum Jülich, Jülich, Germany

**Correspondence:** Lars Hoffmann (l.hoffmann@fz-juelich.de)

**Abstract.** Lagrangian particle dispersion models are indispensable tools for the study of atmospheric transport processes. However, Lagrangian transport simulations can become numerically expensive when large numbers of air parcels are involved. To accelerate these simulations, we made considerable efforts to port the Massive-Parallel Trajectory Calculations (MPTRAC) model to graphics processing units (GPUs). Here we discuss performance optimizations of the major bottleneck of the GPU code of MPTRAC, the advection kernel. Timeline, roofline, and memory analyses of the baseline GPU code revealed that the application is memory-bound and performance suffers from near-random memory access patterns. By changing the data structure of the horizontal wind and vertical velocity fields of the global meteorological data driving the simulations from Structure of Arrays (SoA) to Array of Structures (AoS), and by introducing a sorting method for better memory alignment of the particle data, performance was greatly improved. We evaluated the performance on NVIDIA A100 GPUs of the Jülich Wizard for European Leadership Science (JUWELS) Booster module at the Jülich Supercomputing Center, Germany. For our largest test case, transport simulations with $10^8$ particles driven by the European Centre for Medium-Range Weather Forecasts (ECMWF) ERA5 reanalysis, we found that the runtime for the full set of physics computations was reduced by 75%, including a reduction of 85% for the advection kernel. In addition to demonstrating the benefits of code optimization for GPUs, we show that the runtime of CPU-only simulations is also improved. For our largest test case, we found a runtime reduction of 34% for the physics computations, including a reduction of 65% for the advection kernel. The code optimizations discussed here bring the MPTRAC model closer to applications on upcoming exascale high performance computing systems, and will also be of interest for optimizing the performance of other models using particle methods.

## 1 Introduction

By enabling the application of cutting-edge techniques, graphics processing units (GPUs) have become a cornerstone of Earth system modeling, greatly accelerating simulations and enabling finer resolutions, longer time spans, more complex physics, and larger ensembles (Govett et al., 2017; Fuhrer et al., 2018; Loft, 2020; Bauer et al., 2021; Wang et al., 2021; Giorgetta et al., 2022). GPUs emerged as a transformative technology due to their unprecedented capacity to accelerate complex computational



tasks. These high-performance hardware devices, originally designed for rendering graphics, have been harnessed for scientific applications due to their massively parallel architecture. In the context of Earth system modeling, which involves intricate simulations of atmospheric, oceanic, and other environmental processes, GPUs provide a significant advantage by enabling researchers to process vast amounts of data and execute computationally intensive algorithms at remarkably faster rates than traditional central processing units (CPUs). Earth system modeling demands simulations with high spatial and temporal resolutions to capture complex interactions and phenomena. GPUs excel at handling such requirements, as they can simultaneously perform numerical calculations across multiple compute elements, effectively tackling the parallel nature of Earth system simulations. This capability translates into accelerated model execution, allowing us to run simulations in considerably shorter time frames. As a result, scientists can explore a broader range of scenarios, refine models, and iterate through simulations more rapidly, enhancing the understanding of complex Earth system dynamics.

Given the importance of GPUs for scientific computing, we devoted considerable effort in recent years to porting the Massive-Parallel Trajectory Calculations (MPTRAC) Lagrangian transport model to GPUs (Hoffmann et al., 2022). Lagrangian transport models such as MPTRAC are primarily used to study atmospheric transport processes in the boundary layer, the free troposphere, and the stratosphere. Commonly used Lagrangian models for research and practical applications today are described by Draxler and Hess (1998), McKenna et al. (2002a,b), Lin et al. (2003), Stohl et al. (2005), Jones et al. (2007), Stein et al. (2015), Sprenger and Wernli (2015), and Pisso et al. (2019). For example, the MPTRAC model has been successfully used to assess the dispersion of volcanic emissions from local to global scale (Heng et al., 2016; Hoffmann et al., 2016; Wu et al., 2017, 2018; Liu et al., 2020; Cai et al., 2022; Liu et al., 2023). In addition, the model has been used to study aerosol transport pathways within the upper troposphere and lower stratosphere (Zhang et al., 2020; Smoydzin and Hoor, 2022; Clemens et al., 2023).

Eulerian models represent fluid flows in the atmosphere based on the flow between regular grid boxes of the model. Lagrangian models represent the transport of trace gases and aerosols based on large sets of air parcel trajectories following the fluid flow. Both approaches have distinct advantages and disadvantages. Lagrangian models are particularly well suited for studying fine-scale structures, filamentary transport, and mixing processes in the atmosphere because their spatial resolution is not inherently limited to the resolution of Eulerian grid boxes, and numerical diffusion is particularly low for these models. Lagrangian transport simulations can be quite expensive because diffusion and mixing are based on applying stochastic perturbations to large sets of air parcel trajectories. However, Lagrangian transport simulations are considered to be particularly amenable to parallelization, since the workload of computing large sets of air parcels can be distributed across many compute elements of a CPU or GPU. This is an embarrassingly parallel computational problem because the air parcel trajectories are computed independently of each other. A specific problem, however, is related to memory access patterns, which we address in the present study. Eulerian models typically have well-structured memory access patterns, with computational loops sequentially accessing aligned memory elements, such as the vertical columns of the meteorological fields over the model grid. In contrast, Lagrangian models exhibit near-random memory access patterns to the meteorological data due to the near-random distribution of air parcels in the atmosphere.



In this study, we aimed to optimize the GPU implementation of the MPTRAC Lagrangian transport model to achieve higher simulation throughput and, by better utilizing the GPU devices, also to improve the energy efficiency of the code. To this end, we performed detailed performance analyses of the MPTRAC GPU code, in particular the model's advection kernel, on the

Jülich Wizard for European Leadership Science (JUWELS) Booster high-performance computing (HPC) system at the Jülich Supercomputing Center, Germany. The performance analyses were conducted using the internal timers of the model as well as the NVIDIA Nsight Systems and Nsight Compute tools. The performance analyses revealed that the baseline code of the model is largely memory-bound and that optimization efforts had to focus on optimizing the memory access patterns and the memory layout of the data structures. Two optimizations significantly improved the performance of the code compared to the baseline

version, not only for the advection kernel but for the entire set of physics calculations of the model. While we focused primarily on optimizing MPTRAC's GPU code, which is considered most relevant for application on future exascale HPC systems, we also found that the optimized code provides performance benefits on conventional CPU-based machines. We expect that the code optimizations found here will be helpful also in further optimizing the performance of other models applying particle methods.

This paper is organized as follows. In Sect. 2, we provide a brief overview on the MPTRAC Lagrangian transport model and the advection kernel that was the focus of our code optimization efforts. In Sect. 3 we describe the test case used to evaluate the code performance optimizations of MPTRAC. This includes a description of the JUWELS Booster HPC system at the Jülich Supercomputing Center on which the tests were performed. Section 4 summarizes the main results of the performance analysis of the baseline and optimized code versions of MPTRAC. In particular, we compare timeline analyses of the simulations and

compare roofline and memory chart analyses of the advection kernel obtained with Nsight Systems and Nsight Compute. In Sect. 5, we discuss the optimization of the meteorological data memory layout and in Sect. 6, we discuss the optimization of the particle data memory layout in more detail. Section 7 summarizes the benefits of the code optimizations for CPU-only simulations. Finally, Sect. 8 summarizes the results of the study and gives our conclusions.

## 2 The MPTRAC Lagrangian transport model

MPTRAC (Hoffmann et al., 2016, 2022) is a Lagrangian particle dispersion model for the analysis of atmospheric transport processes in the free troposphere and the stratosphere. MPTRAC calculates the trajectories of air parcels by solving the kinematic equation of motion using given horizontal wind and vertical velocity fields from global reanalyses or forecasts. Eddy diffusion and subgrid-scale wind fluctuations are simulated applying the Langevin equation to add stochastic perturbations to the trajectories. Additional modules are implemented to simulate convection, sedimentation, exponential decay, gas and aque-

ous phase chemistry, and wet and dry deposition. The meteorological data pre-processing code of MPTRAC provides estimates of boundary layer height, convective available potential energy, geopotential heights, potential vorticity, and tropopause data. Different types of model output are available, including particle, grid, ensemble, profile, sample, and station data. The output can be written in different data formats, including the Network Common Data Format (netCDF) as well as data files for the



Gnuplot and ParaView visualization tools. The code features a hybrid MPI-OpenMP-OpenACC parallelization for efficient use
from single workstations up to HPC and GPU clusters.

Figure 1 shows an overview of the geophysical modules and other software components of MPTRAC. Here we roughly
divided the geophysical modules into the modules that deal with the trajectory calculations and the modules that deal with the
chemistry calculations. The former modules modify the positions of the air parcels, while the latter modules affect the trace
gas or aerosol concentrations assigned to them. Meteorological input data can be ingested in a variety of data formats, the most
common being netCDF. The meteorological data pre-processing code allows the user to derive various diagnostic variables
such as geopotential heights, potential vorticity, or tropopause data from the primary state variables of the meteorological input
data. Model output can be written in various types, with particle and grid output being most commonly used. A set of utility or
infrastructure components is available for basic tasks such as interpolation of the meteorological data. In this study, we revised
and optimized the code for handling the meteorological data and added a new module for sorting the particle data in memory to
optimize the memory access patterns and thereby the performance and energy-efficiency of Lagrangian transport simulations
with MPTRAC on GPUs.

Our optimization efforts focused largely on MPTRAC's advection kernel, which was found to consume most of the runtime
for large simulations. The advection kernel provides the numerical solution to the trajectory equation,

$$\frac{d\boldsymbol{x}}{dt} = \boldsymbol{v}(\boldsymbol{x}, t), \tag{1}$$

where $\boldsymbol{x}(t)$ denotes the position of an air parcel at time $t$ in response to a given wind and velocity field $\boldsymbol{v}(\boldsymbol{x}, t)$. The geographical
coordinate system $\boldsymbol{x} = (\lambda, \phi, p)$ specifies the position with longitude $\lambda$, latitude $\phi$ and pressure $p$. The velocity vector $\boldsymbol{v} = (u, v, \omega)$ contains the zonal wind component $u$, the meridional wind component $v$, and the vertical velocity $\omega$. The explicit
midpoint method is used to solve the trajectory equation, expressed as

$$\boldsymbol{x}(t + \Delta t) = \boldsymbol{x}(t) + \Delta t\, \boldsymbol{v}\left\{\boldsymbol{x}(t) + \frac{\Delta t}{2}\boldsymbol{v}\left[\boldsymbol{x}(t), t\right], t + \frac{\Delta t}{2}\right\}. \tag{2}$$

This method is known for its balanced trade-off between accuracy and computational efficiency (Rößler et al., 2018). Following
the common practice in Lagrangian transport models (Bowman et al., 2013), 4-D linear interpolation is applied to the provided
wind and velocity fields in both spatial and temporal dimensions.

Lagrangian transport simulations with MPTRAC are driven by global meteorological reanalyses or forecasts, including vari-
ous data products provided by the European Centre for Medium-Range Weather Forecasts (ECMWF), the National Aeronautics
and Space Administration (NASA), the National Centers for Environmental Prediction (NCEP), and the National Center for
Atmospheric Research (NCAR). In this study, we use the ECMWF ERA5 (Hersbach et al., 2020) and ERA-Interim (Dee et al.,
2011) reanalyses to drive the transport simulation and conduct the performance analyses of MPTRAC. While ERA5, launched
in 2016, represents the current state of global meteorological reanalysis at ECMWF, its predecessor ERA-Interim, launched in
2006, is considered here mainly for comparison because it has a much lower data volume and spatiotemporal resolution than
ERA5. Specifically, ERA5 provides hourly meteorological fields at $T_L 639$ ($\sim 31\,\mathrm{km}$) horizontal resolution on 137 vertical
levels from the surface up to 0.01 hPa. ERA-Interim provides 6-hourly fields at $T_L 255$ ($\sim 79\,\mathrm{km}$) horizontal resolution on 60
levels up to 0.1 hPa.



**Figure 1.** Overview of the geophysical modules and other software components of the Lagrangian transport model MPTRAC.

Handling the large volume of ERA5 data in Lagrangian transport simulations is a particular challenge. For example, when conducting 10-day forward trajectory calculations on the Jülich Research on Exascale Cluster Architectures (JURECA) HPC

system at the Jülich Supercomputing Center, Hoffmann et al. (2019) found that calculations with ERA5 require about $10\times$ more runtime and main memory and about $80\times$ more disk space than the corresponding simulations with ERA-Interim. Comparing transport simulations with ERA5 and ERA-Interim in this study allows us to evaluate our code optimizations and the performance of MPTRAC with respect to different size and resolution of the input data. For further details on how we retrieved the reanalysis data from ECMWF and how they are pre-processed for use with MPTRAC, see Hoffmann et al. (2022).





## 3 Description of the test case and environment

We conducted detailed performance benchmarking with the baseline and optimized code versions of MPTRAC. In the test case for the transport simulations considered here, we used the modules for simulating the advection of air parcels, the eddy diffusion and subgrid-scale wind parameterizations, and the extreme convection parameterization. We considered loss processes due to an exponential decay of particle mass, chemical reaction with hydroxyl, and wet deposition. We did not limit the test case for benchmarking to advection alone in order to assess how code optimizations affect also other parts of the code.

The transport simulations were initialized on 1 January 2017, 00:00 UTC and cover a period of 24 h. Two different sets of meteorological data are considered to assess how performance scales with the size of the input data. The simulations were driven by hourly ERA5 data with a model time step of 180 s or 6-hourly ERA-Interim data with a model time step of 360 s. The runtime required for model output varies greatly depending on the selection of the type and frequency of the output. We have chosen two types of output as representative examples. Particle data output, comprising individual data such as the position and additional variables per air parcel, was written every 6 hours. Grid output, comprising mean or aggregated air parcel data over regular grid boxes, was written every hour.

We used sets of $10^5$ to $10^8$ globally distributed air parcels to see how performance scales with problem size. The random horizontal distribution of the air parcels was scaled by the cosine of latitude to achieve nearly homogeneous horizontal coverage. For the vertical distribution, we used a uniform random distribution from the surface up to $\sim$60 km altitude. A set of globally distributed air parcels was chosen here for testing, rather than a test case representing the spread of emissions from a point source. We have chosen globally distributed air parcels to ensure that most of the meteorological data is accessed as input during the simulations. As a larger amount of data needs to be handled, this is technically more challenging than a test case using only a subset of the meteorological data in a local case study.

Performance benchmarking was conducted using version 2.6 of MPTRAC. Both, the CPU and GPU versions of MPTRAC were compiled using the NVIDIA HPC Software Development Kit (SDK) C compiler, version 23.1. Common optimization flags have been applied, see the Makefile located in the code repository for more details. Note that data transfers between the host (CPU) and the device (GPU) memory are explicitly specified in the code rather than using the NVIDIA Unified Memory feature. To optimize OpenMP scaling of the compute jobs, process binding was enabled (`OMP_PROC_BIND=true`), i. e., threads remain pinned to cores throughout the simulation. The threads were placed on individual sockets (`OMP_PLACES=sockets`). We disabled simultaneous multithreading (via the Slurm scheduler with `--hint=nomultithread`), i. e., the number of compute tasks and logical cores corresponds to the number of physical cores.

The Booster module of JUWELS (Jülich Supercomputing Centre, 2019, 2021, 2023) was employed for all calculations. The machine consists of 936 compute nodes. Each node is equipped with two AMD EPYC Rome 7402 CPUs and four NVIDIA A100 GPUs connected to other GPUs via 200 GByte/s third generation NVLink. Each node has four InfiniBand HDR 200 adapters. Nodes are connected to each other in a DragonFly+ topology. A total of 3744 A100 GPUs put the JUWELS Booster in 13th place on the Top500 list, even three years after its installation. Table 1 summarizes technical specifications of the JUWELS Booster compute nodes.





**Table 1.** Technical specifications of JUWELS Booster compute nodes

| | |
|---|---|
| **CPUs (2 per node)** | |
| CPU model | AMD EPYC Rome 7402 |
| CPU architecture | AMD Zen 2 |
| Base frequency | 2.8 GHz |
| CPU Memory | 512 GByte DDR4, 3200 MHz |
| Memory bandwidth | 204.8 GByte/s (per socket) |
| Cores | 24 physical / 48 virtual (SMT) |
| Host to device interface | PCIe Gen4 x128 (64 GByte/s) |
| Thermal design power | 180 W |
| **GPUs (4 per node)** | |
| GPU model | NVIDIA A100 40GB SXM4 |
| GPU architecture | NVIDIA Ampere GPU architecture (CC 8.0) |
| FP64 performance | 9.7 (19.5 tensor core) TFLOP/s |
| FP32 performance | 19.5 (156 TF32) TFLOP/s |
| GPU memory | 40 GByte HBM2 |
| Memory bandwidth | 1.6 TByte/s |
| Cores | 3456 (FP64) / 6912 (FP32) |
| GPU to GPU interface | Third generation NVIDIA NVLink (600 GByte/s) |
| Thermal design power | 400 W |

## 4 Performance analysis of baseline and optimized code

In this section, we discuss performance analyses of the original code of MPTRAC version 2.6, also referred to as the baseline code, and optimized versions of the model, which include optimizations of the memory layout and the memory access pattern of the meteorological data and the particle data. The details of the code improvements are discussed further in Sects. 5 and 6. We first performed a timeline analysis using Nsight Systems, which mainly helped us to identify the performance bottlenecks and hotspots of the code that needed further investigation. This analysis showed that the advection kernel of MPTRAC was 170 the major bottleneck of the baseline code. Next, we used Nsight Compute to perform a roofline and memory analysis of the advection kernel to provide further guidance on how to optimize the code. The performance analysis with Nsight Systems and Nsight Compute is illustrated here for the largest test case considered in the study, consisting of a set of $10^8$ particles driven by ERA5 meteorological fields.

Figure 2 shows screenshots of the timeline analysis with Nsight Systems for the baseline and optimized code. We selected 175 a single time step of the MPTRAC time loop for illustration. The plots show timelines as stacked rows for, from top to bottom, GPU utilization of the code, a selection of the most relevant compute kernels of MPTRAC, the NVTX markers




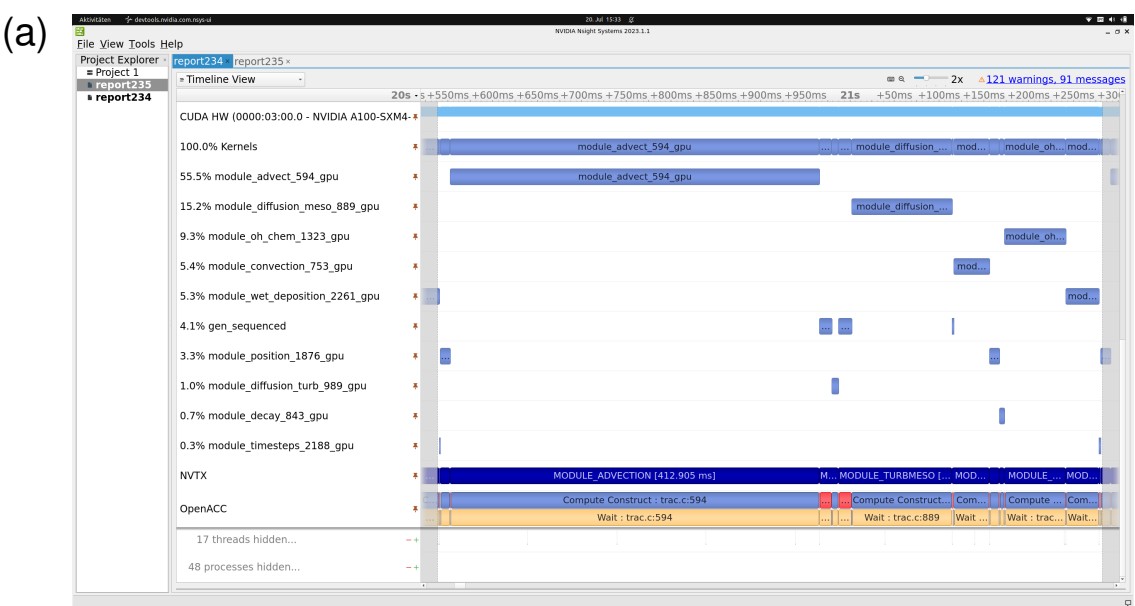

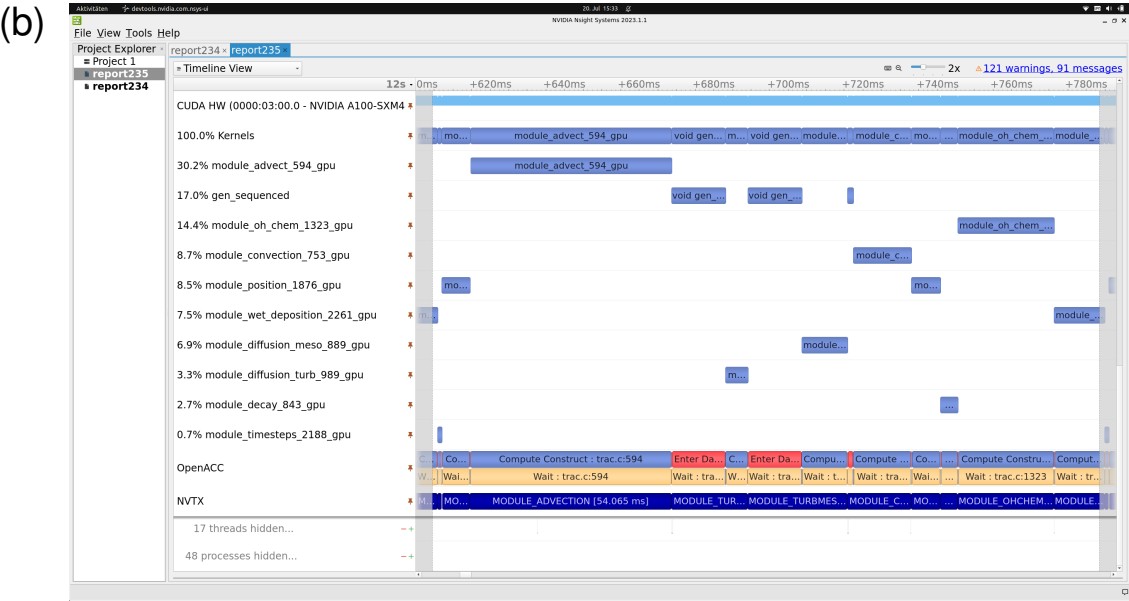

**Figure 2.** Timeline analyses of a single time step of the MPTRAC Lagrangian transport model for a simulation with $10^8$ globally distributed air parcels driven by ERA5 data. Results are shown for (a) the baseline code of MPTRAC and (b) an optimized code version with improved data structures and memory access patterns. The analyses were conducted on the JUWELS Booster HPC system at the Jülich Supercomputing Centre, Germany. The Nsight System report files used to create this figure are made available in the electronic supplement to the paper.




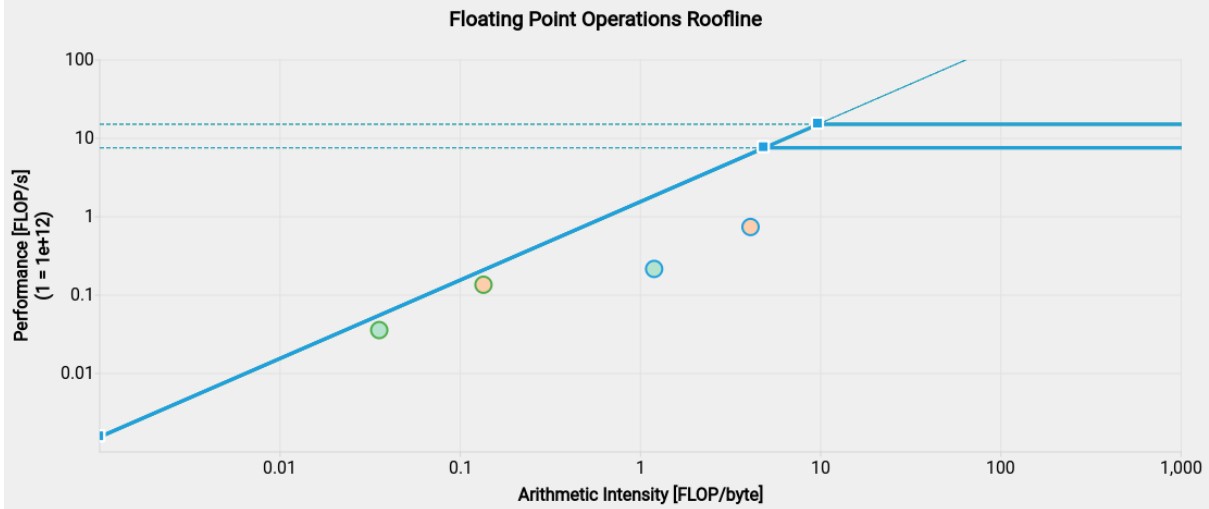

**Figure 3.** Roofline analysis of the baseline (green and orange points) and optimized (blue and red points) version of the advection kernel of the MPTRAC Lagrangian transport model. Results are shown for single precision (green and blue points) and double precision (orange and red points) calculations, respectively. Constant lines indicate the compute performance limits of the NVIDIA A100 GPU devices for double and single precision, respectively. The slanted line to the left indicates the memory bandwidth limit of the GPU device. The Nsight Compute report files used to create this figure are made available in the electronic supplement to the paper.

placed throughout the code to indicate the different modules and code sections, and the OpenACC compute constructs and data transfers that are performed. The runtime required to compute a single time step is about 740 ms for the baseline code (Fig. 2a). In the baseline code, the advection kernel consumes about 55% of the runtime, followed by subgrid-scale wind fluctuations (15%), hydroxyl chemistry (9%), convection (5%), wet deposition (5%), random number generation (4%), and air parcel position checking (3%). Other modules of MPTRAC used in the test case require 1% or less of the runtime of a time step.

Since the advection kernel consumes most of the runtime of the baseline code, we decided to further analyze and optimize this part of the code, as discussed below. A time step of the fully optimized code requires about 180 ms (Fig. 2b), a reduction of 76% compared to the baseline code. The distribution of the runtime between the different modules is more balanced in the optimized code. The advection kernel consumes about 30% of the runtime of a time step, followed by random number generation (17%), hydroxyl chemistry (14%), convection (9%), air parcel position checking (9%), wet deposition (8%), subgrid-scale wind fluctuations (7%), eddy diffusion (3%), and exponential decay (3%). As pointed out earlier, the runtime was not only better balanced between the different modules of the optimized code, but also significantly reduced overall. It is important to note that the optimizations we introduced did not only benefit the advection kernel, but also other parts of the code.

Figure 3 shows the results of the roofline analysis of MPTRAC's advection kernel as obtained with Nsight Compute. The roofline model visualizes floating point performance (in units of FLOP/s) as a function of arithmetic intensity (in units of FLOP/byte). It represents the theoretical performance bounds below which kernel or application performance exists. The



roofline model is defined by two platform-specific performance bounds, i. e., a bound derived from memory bandwidth and a
bound derived from processor peak performance. The roofline analysis for the advection kernel shows that the code is memory-
bound, i. e., it spends most of its time accessing memory rather than computing.

   Specifically, the roofline analysis shows that the baseline code had a compute performance of 131.5 GFLOP/s and an arith-
metic intensity of 0.14 FLOP/byte for double precision and a performance of 34.7 GFLOP/s and an arithmetic intensity of
0.04 FLOP/byte for single precision. Our code improvements increased the performance to 716.8 GFLOP/s and an arith-
metic intensity of 4.14 FLOP/byte for double precision and a performance of 209.2 GFLOP/s and an arithmetic intensity of
1.21 FLOP/byte for single precision. This is an increase in performance of a factor of 5.5 for double precision and 6.0 for single
precision. Note that most of the computations in MPTRAC's advection kernel are conducted in double precision. Note also that
the peak floating point performance of the NVIDIA A100 GPU devices (Table 1) is much higher than the performance values
found here. Peak performance values are theoretical limits that can only be approached by highly optimized compute-bound
algorithms, whereas the advection kernel of MPTRAC is memory-bound.

   Figure 4 shows a memory chart for the baseline and optimized code of the MPTRAC advection kernel as obtained with
Nsight Compute. This graph shows that the baseline code (Fig. 4a) has relatively low hit rates of 45.56% for accessing the
L1/TEX cache and 52.71% for accessing the L2 cache. The throughput from the GPU device memory to the L2 cache is at
a level of 850.71 GByte/s, which exceeds 60% of the peak bandwidth. The Nsight Compute analysis suggests that the main
issue of the baseline code of the advection kernel arises from its memory access patterns. The access patterns of the advection
kernel are nearly random due to the fact that the air parcels are randomly distributed on the global scale in our simulation.
Sequential access patterns, where data is processed with straightforward incremented or decremented addressing, are highly
amenable to prefetching. In contrast, near-random access patterns break the principle of locality and usually inhibit performance
optimization through the use of techniques such as the caching.

Large global meteorological fields from state-of-the-art reanalyses such as ERA5 require a significant amount of device
memory of the GPU. For example, storing ERA5 data for two time steps for use with MPTRAC requires about 14 of the
40 GByte device memory of the JUWELS Booster's NVIDIA A100 GPUs. If we access this data in a nearly random manner,
there is a rather small chance that it is available directly from the L1/TEX cache (having a size of 192 kByte) or from the L2
cache (40 MByte). Accessing the data requires expensive data transfers from device memory through the cache hierarchy. From
the performance analysis of the baseline code, we concluded that optimization efforts had to focus primarily on optimizing data
structures and memory access patterns. For reference, Fig. 4b shows the memory chart of the optimized code. The hit rates
increased to 81.36% for the L1/TEX cache and 85.85% for the L2 cache. The throughput from GPU device memory to the L2
cache was reduced to 122.15 GByte/s, which is less than 10% of the peak bandwidth. The code optimizations we applied to
achieve these improvements are discussed in more detail in the following sections.





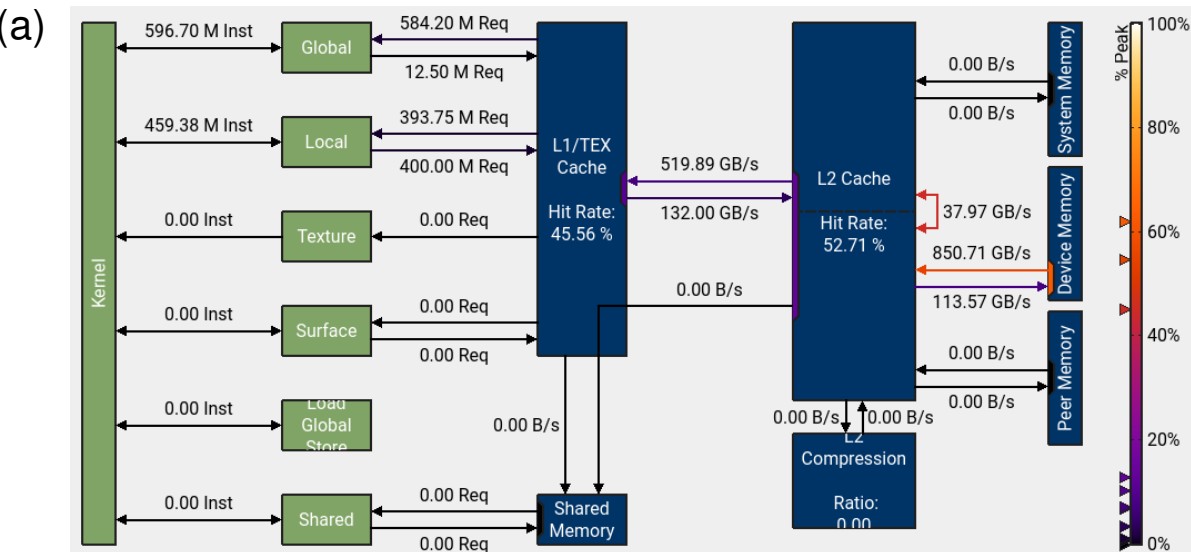

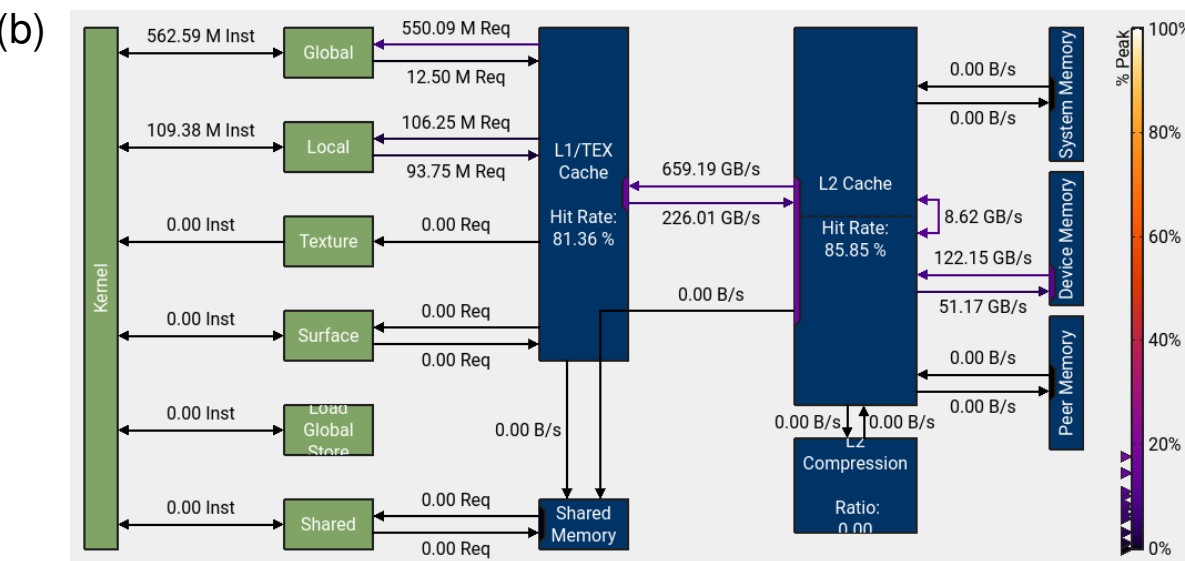

**Figure 4.** Memory chart for the baseline (a) and optimized (b) code of the MPTRAC advection kernel for the largest ERA5 test case. Green boxes show the memory elements directly connected to the GPU's compute elements. Blue boxes show the L1/TEX and L2 caches, device memory, etc. The hit rates indicate how often data is accessible directly from the caches. Labels along the arrows indicate the throughput (in units of GByte/s) within the memory architecture. Color coding indicate the fraction of peak bandwidth. The Nsight Compute report files used to create this figure are made available in the electronic supplement to the paper.





## 5 Optimization of meteorological data memory layout


In this study, we identified two methods for accelerating Lagrangian transport simulations with MPTRAC. Both methods involve optimizing the memory layout of the code's data structures. The first method optimizes the data structures of the meteorological fields, in particular of the horizontal wind and vertical velocity components $(u, v, \omega)$. In the baseline code, the wind and velocity components are stored in three separate 3-D arrays as floats with single precision, declared in the C

language as `float u[EX][EY][EZ], v[EX][EY][EZ], w[EX][EY][EZ]`. The size of the 3-D arrays is specified by the number of longitudes, `EX`, the number of latitudes, `EY`, and the number of pressure levels, `EZ`. In the C programming language, arrays are arranged in memory in row order, as opposed to Fortran, where they are arranged in column order. In our case, the elements along the `EZ` dimension are most closely aligned in memory, followed by the `EY` dimension, followed by the `EX` dimension.

The memory layout chosen in the baseline code of MPTRAC is best suited for computational kernels that operate on individual vertical profiles of the meteorological data, since the data of individual vertical profiles are aligned and can be effectively transferred in the form of cache lines from main memory to the memory caches of the GPU or CPU during program execution. This property is exploited in MPTRAC's meteorological data pre-processing code, which is used to compute various diagnostic meteorological variables such as geopotential heights or potential vorticity from vertical profiles of basic state variables such as

pressure, temperature, humidity, or horizontal winds and vertical velocity as provided by the external meteorological model. In contrast, compute kernels that operate on individual horizontal levels of the data are less efficient with the given data structure because the data are spread more widely across memory, causing unnecessary data transfers of cache lines from main memory to caches where only few elements actually need to be accessed. Time-consuming cache misses occur more frequently when there is frequent and uncoalesced access to data outside the cache lines.

The memory layout of the data structures and the access patterns of the compute kernels therefore have to be carefully considered to optimize memory access. MPTRAC's advection kernel operates in an outer loop over all air parcels, which is either distributed across the CPU's compute cores via OpenMP or offloaded to the GPU via OpenACC. Inside the loop, the kernel applies the mid-point scheme for numerical integration to determine the trajectories of the air parcels. The mid-point scheme requires 4-D linear interpolation of the wind and velocity components in space and time. Since the wind and velocity

components are needed in immediate succession to compute the trajectories, we rearranged and merged the 3-D arrays of the wind and velocity components into a single 4-D array so that the wind and velocity components are directly aligned in memory. The array is declared as `float uvw[EX][EY][EZ][3]`. In this revised data structure, the innermost index of the array `uvw` refers to one of the three wind and velocity components, respectively. Technically, this resembles a change of data structures from Structure of Arrays (SoA) of the separate arrays `u`, `v`, and `w` to Array of Structures (AoS) for the combined

array `uvw`.

Figure 5 shows selected runtime measurements for different parts of the MPTRAC model for our largest test case, using ERA5 data for input and considering the maximum of $10^8$ particles in this study. By optimizing the memory layout of the wind and velocity fields, the total runtime of the simulations (Fig. 5a) was reduced from 477 to 365 s ($-23\%$). The physics





runtime was reduced from 377 to 265 s ($-30\%$). The runtime for file I/O and data transfers between CPU and GPU memory

did not change notably, as expected. A breakdown of the physics timers (Fig. 5b) shows that the runtime of the advection kernel was reduced from 189 to 105 s ($-44\%$) and that of the subgrid-scale wind parameterization from 61 to 32 s ($-48\%$). The other geophysical modules of MPTRAC did not show any improvements, which was expected since they do not access the optimized wind and velocity fields. However, the optimizations of the advection kernel and the parameterizations of the subgrid-scale wind fluctuations alone yield significant overall improvements in the physics computations and total runtime of

the simulation.

Figure 6 shows the scaling of the runtime and the speedup of the optimized code (labeled 'Meteo fields', green bars) with respect to the baseline. Since the total runtime of the code is affected by various aspects, we focus here on the physics calculations covering the selected computational kernels included in the test case and the specific results for the advection kernel. The scaling results are shown for different problem sizes, represented by the number of particles varying from $10^5$ to

$10^8$. The tests were performed using both ERA5 and ERA-Interim fields to drive the transport simulations, in order to assess the dependence on the size of the input data. The log-log plots of runtime (Fig. 6a, c) show nearly ideal, linear scaling for large problem sizes ($\geq 10^6$ particles) and only slight degradations from linear scaling for smaller problem sizes. This nearly ideal scaling with respect to problem size is observed for both ERA5 and ERA-Interim data. The estimated speedup of the optimized code over the baseline code (Fig. 6b, d) is mostly greater than one, indicating improvements in runtime. The speedup due to

the optimization of the meteorological data layout increases with increasing numbers of particles, but tends to saturate for $10^6$ particles or more. For the full set of physics kernels, we found a maximum speedup of 1.4 for both ERA5 and ERA-Interim. For the advection kernel, we found a maximum speedup of 1.8 for ERA5 and a maximum speedup of 2.0 for ERA-Interim.

We performed an additional experiment on the data structures of the meteorological fields, testing whether including the temperature ($T$) as a fourth component of the AoS in addition to the wind and velocity fields ($u, v, \omega$), resulting in the combined

4-D array `float uvwt[EX][EY][EZ][4]`, would bring further runtime benefits due to vectorized memory access. The test showed that the performance of the advection module with the 4-D array including temperature was lower, with the runtime of the ERA5 simulations with $10^8$ particles increasing by 20% and ERA-Interim by 3%. Thus, on GPUs, further expansion of the AoS structure may only be worthwhile if computationally intensive modules would be added that combine the use of the wind fields with temperature in a single computational kernel. This might be relevant in future work, as we intend to implement

diabatic vertical transport in MPTRAC.

## 6 Optimization of particle data memory layout

The memory analysis with Nsight Compute showed that the advection kernel is memory-bound and that the runtime depends largely on the data transfers of meteorological data from the GPU main memory to the caches and registers of the GPU compute elements (Sect. 4). Starting from a set of near-random globally distributed particles, the corresponding memory access patterns

of the advection kernel are also near-random. This leads to a notable fraction of cache misses and a slowdown in the overall




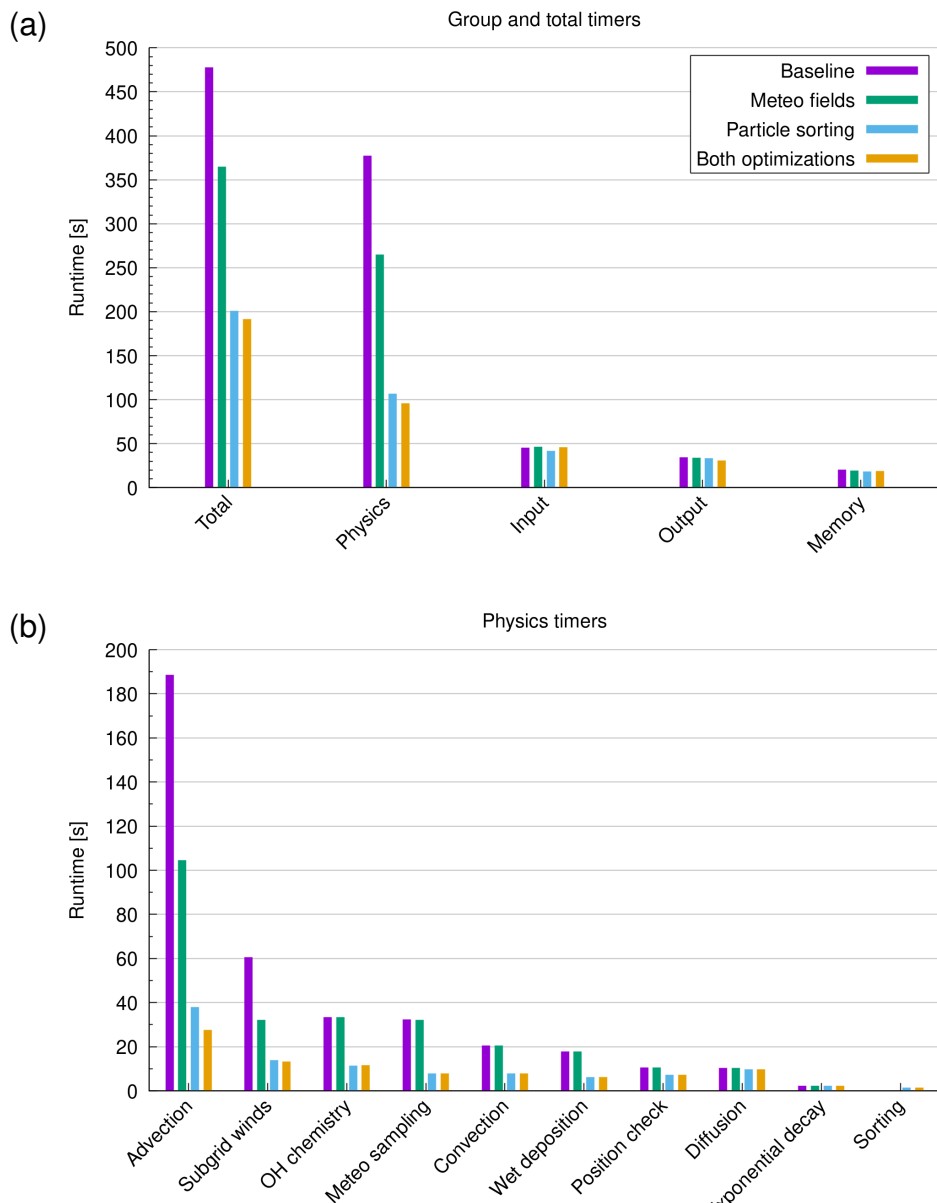

**Figure 5.** Runtime measurements for the largest test case using $10^8$ particles and ERA5 data as input. Panel (a) shows the total runtime and the group timers for physics computation, file I/O, and data transfers between CPU and GPU memory. Panel (b) shows a breakdown of the physics timers. Results are shown for the baseline code, for the optimization of the memory layout of the wind and velocity fields, for the sorting of the particle data, and for the full optimization including both methods (see plot key).

runtime of the kernel. Significant improvements in the runtime of the compute kernel were expected due to better alignment of memory access.





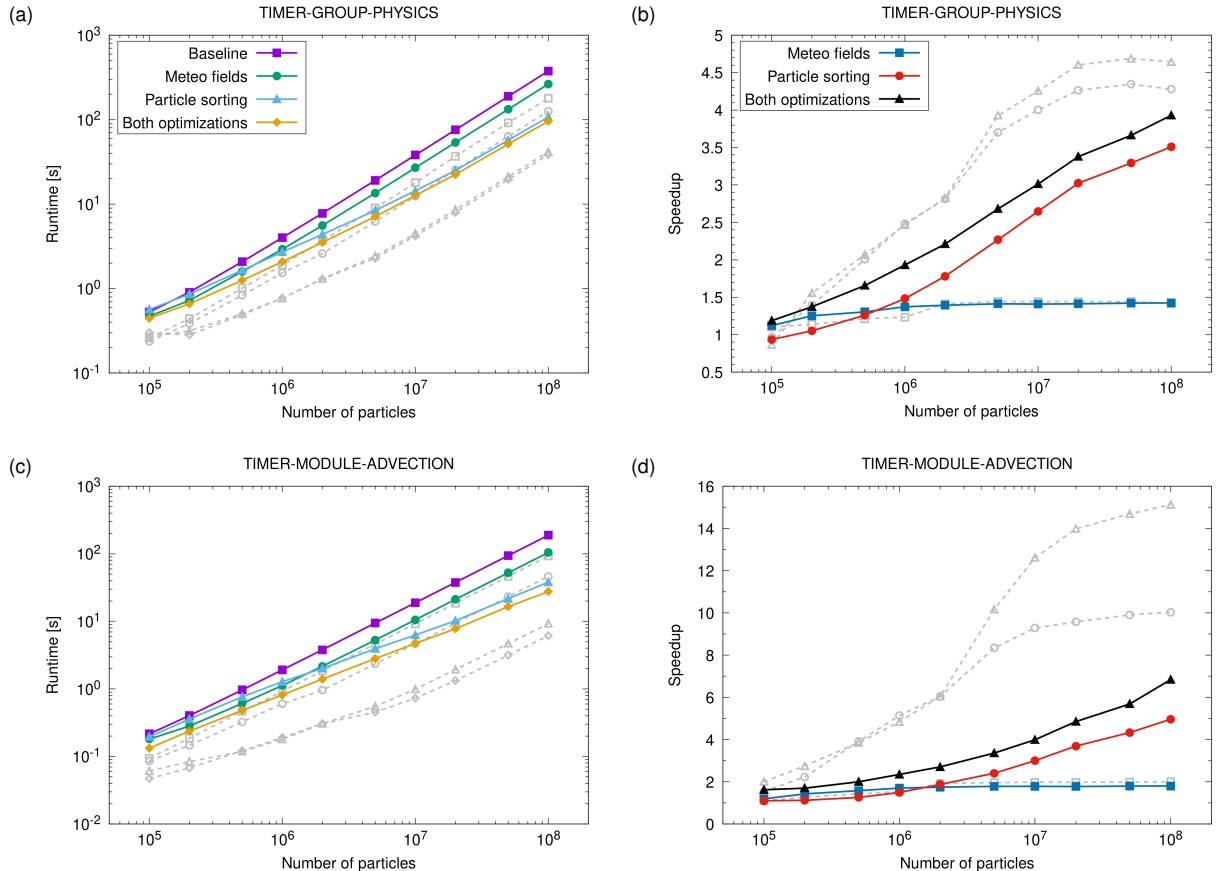

**Figure 6.** Runtime (a, c) and speedup (b, d) of the optimized code versions with respect to the baseline code of MPTRAC (see plot key). Scaling results for the full set of physics kernels (a, b) and the advection kernel only (c, d) are shown for different numbers of particles. Colored curves show results for ERA5 data. Gray, dashed curves show the corresponding results for ERA-Interim.

For this reason, we utilized a sorting algorithm that reorders the particle data in memory to better align memory access. MPTRAC's new sorting module consists of three steps. In the first step, we compute a linear box index for each air parcel that 295 represents the target location in memory. For an air parcel located in the longitude-latitude-pressure box $[\lambda_i, \lambda_{i+1}] \times [\phi_j, \phi_{j+1}] \times [p_k, p_{k+1}]$, the linear box index is calculated as $l = (i \times n_{lat} + j) \times n_p + k$, where $n_{lat}$ and $n_p$ are the numbers of latitudes and pressure levels of the 3-D meteo fields, respectively. Considering that the 3-D fields are stored in row-major order in our code, sorting the particles data according to the linear box index $l$ ensures that we access the meteorological data in order when we linearly iterate over all the particles. In the second step, we therefore apply a sorting algorithm to find the permutation $P$ of the 300 particles that will reorder them according to the box index $l$. Here we used the key-value sorting function of the Thrust parallel algorithms library (Bell and Hoberock, 2012), which provides efficient implementations of sorting algorithms for both, CPUs and GPUs. For primitive data types such as integer numbers, Thrust dispatches a highly-tuned radix sort algorithm which is considerably faster than alternative comparison-based sorting algorithms such as merge sort. In the third step, the individual





arrays of particle data, i. e., their positions and any additional variables per air parcel, are sorted according to $P$. All three steps are parallelized using CUDA and OpenACC for offloading to GPUs or OpenMP for executing on CPUs.

Figure 5 shows that there are significant runtime improvements due to particle sorting for the largest test case using ERA5 data and $10^8$ particles. The total runtime of the simulations (Fig. 5a) was reduced from 477 to 201 s ($-58\%$). The physics runtime was reduced from 377 to 107 s ($-72\%$). The breakdown of the physics timers (Fig. 5b) shows that not only the runtime of the advection kernel, but also of most of the other compute kernels was reduced. The largest reductions in runtime were found for the advection kernel ($-80\%$), the subgrid-scale wind fluctuations ($-77\%$), the sampling of meteorological data along the trajectories ($-76\%$), the hydroxyl chemistry ($-66\%$), and wet deposition ($-65\%$). These compute kernels require frequent access to the meteorological fields. The smallest reductions were found for the calculation of exponential decay ($-1\%$) and eddy diffusion ($-6\%$). These kernels do not require access to the meteorological data and therefore do not benefit from realignment. Note that code outside of the physics calculations can also benefit from particle sorting. For example, the runtime required to write the grid output is reduced because the grid output requires the calculation of box averages of the particle data, which is faster if the particle data are already sorted according to the grid boxes of the meteorological data.

The results of the scaling test in Fig. 6 show that the speedup due to particle sorting increases with the number of particles. This was expected, since as the number of particles increases, there is a greater chance that the sorted particle data will allow better reuse of the meteorological data from the caches. In particular, if there are multiple particles in the same grid box of the meteorological data, this will allow very efficient reuse of the data from the caches. For the physics timers, we found a maximum speedup of a factor of 3.5 for ERA5 and 4.4 for ERA-Interim for $10^8$ particles from particle sorting over the baseline code. For the advection kernel, we found a speedup of a factor of 5.0 for ERA5 and 10.0 for ERA-Interim. ERA-Interim has much larger grid boxes than ERA5 and a smaller data volume, which means that there is a higher chance that particles are in the same or neighboring grid boxes and can effectively reuse data from the caches, resulting in a much higher speedup for ERA-Interim than ERA5. The 3-D fields of the ERA5 data cover about $98.9 \times 10^6$ grid cells, while the ERA-Interim data cover about $6.8 \times 10^6$ grid cells, which is a factor of 14.5. Possibly, the remarkable increase in speedup for ERA-Interim for $5 \times 10^6$ particles or more is due to the fact that the average number of particles per grid cell of the meteorological data exceeds a factor of one at this point.

Since sorting can be numerically expensive, we introduced a control parameter that allows the user to select the time interval $\Delta t_s$ for sorting. Reducing the sorting frequency takes advantage of the fact that many particles will remain close to properly sorted for several time steps, and will only eventually become unsorted due to advection, diffusion, and other processes as the simulation progresses. This is illustrated in Fig. 7, which shows individual runtime measurements of the advection kernel during the simulation for sorted and unsorted particle data. For illustration, the particles were sorted only once at the beginning of the simulations in this example. During the 10-day time period shown here, it is found that the particles become unsorted much faster in the ERA5 simulation than in the ERA-Interim simulation. This is attributed to the fact that ERA5 has smaller grid boxes as well as grid scale variability than ERA-Interim.

For the performance measurements presented earlier, we have chosen $\Delta t_s = 1$ h for sorting, which corresponds to 20 time steps of the simulation with the ERA5 data. With this setting, sorting causes a rather small overhead of only 1.5 s or 1.4% of

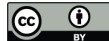



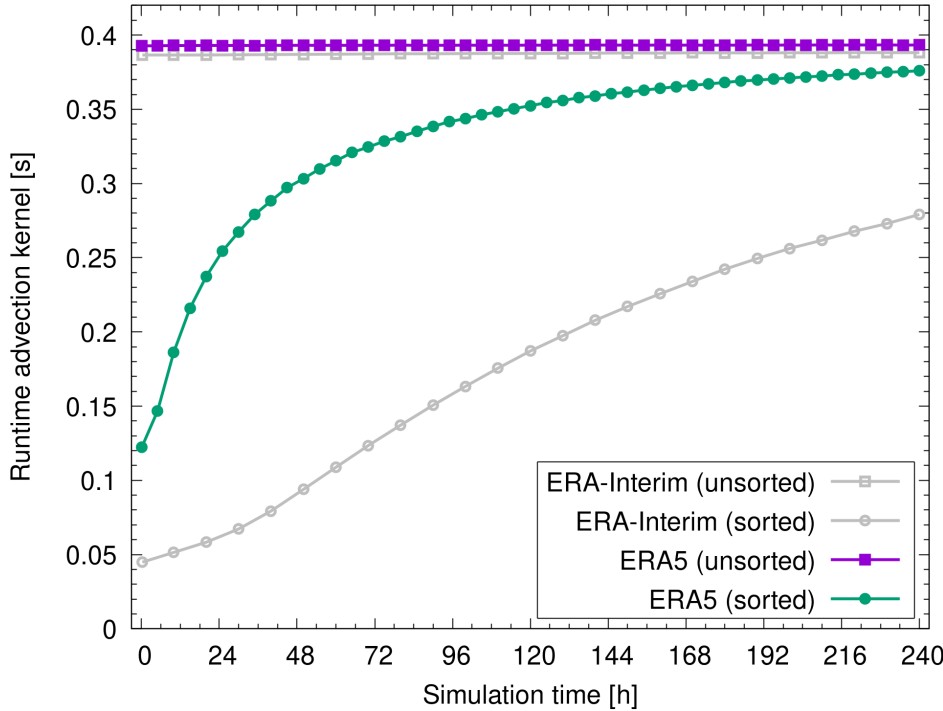

**Figure 7.** Runtime measurements of the advection kernel during a 10-day time period of ERA5 and ERA-Interim simulations with $10^8$ particles. Different curves show the runtime measurements for unsorted particle data and for sorting the particles once at the beginning of the simulations (see plot key).

the physics runtime for the largest ERA5 test case on the GPUs in total. This surprisingly small overhead is due to the highly
efficient GPU implementation of the Thrust library's key-value sorting algorithm. In general, there is a trade-off to consider, as
more frequent sorting reduces the physics runtime, but requires more time for the sorting itself, and vice versa. We therefore
performed a parameter study on the choice of $\Delta t_s$ to find the best trade-off regarding the sorting frequency. Figure 8 shows
the speedup of the set of physics timers, including the particle sorting, for different choices of $\Delta t_s$. For the largest simulations
with $10^8$ particles, it confirms that the best speedup is achieved for $\Delta t_s = 1\,\text{h}$ for ERA5, whereas $\Delta t_s = 2\,\text{h}$ is recommended
for ERA-Interim.

Finally, we evaluated the benefits of combining both optimization methods, the improved memory layout of the meteorological fields (Sect. 5) and the particle data sorting (this section). Figures 5 and 6 show that most of the runtime improvements
are actually due to particle sorting but adding the optimization of the memory layout of the meteorological fields still provides
additional improvements. For the largest ERA5 test case with $10^8$ particles, the total runtime was reduced by another 5% for
the full optimizations compared to particle sorting alone (Fig. 5). The physics runtime was reduced by 10% and the advection
kernel runtime was reduced by 27%. The maximum speedup increased to 2.5 in total runtime, 3.9 for the physics runtime, and
6.8 for the advection kernel (Fig. 6b, d) compared to the baseline.



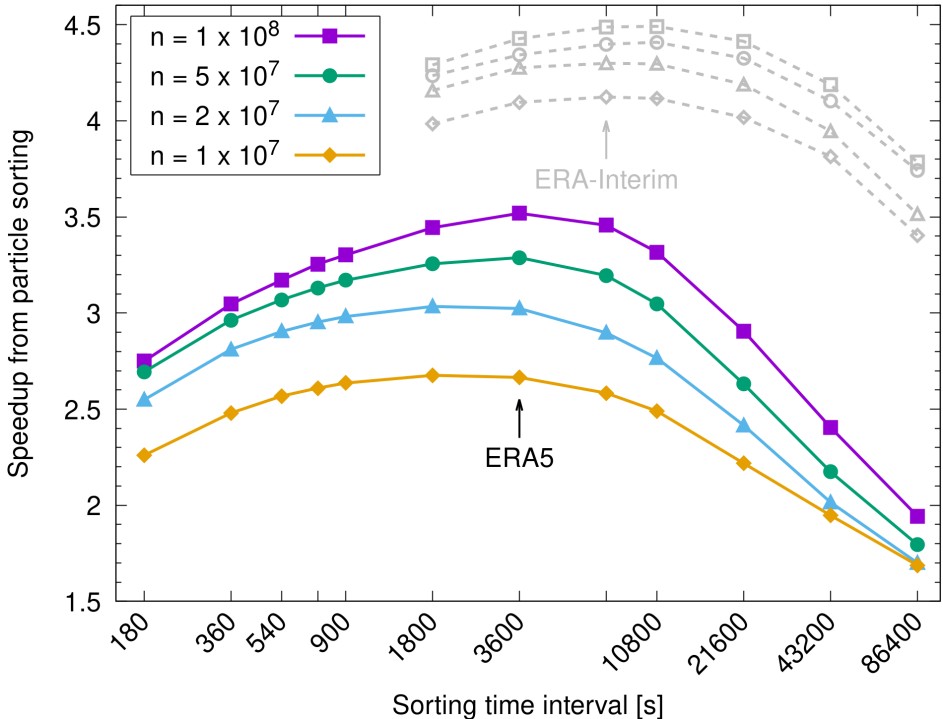

**Figure 8.** Speedup of physics calculations due to particle sorting for different sorting time intervals $\Delta t_s$. Colored curves show results for ERA5 data for different numbers of particles $n$ (see plot key). Gray curves show the corresponding results for ERA-Interim.

## 7    Assessment of code optimization for CPUs

Throughout this study, we primarily focused on GPU optimizations of the MPTRAC model. In this section, we discuss whether
the code optimizations also led to improvements when running on CPUs. Note that the codebase of the CPU and GPU versions
of MPTRAC is mostly the same, except that OpenMP is used to distribute the computations in the particle loop across the
CPU cores, whereas OpenACC is used to offload and parallelize the loop on the GPU device. In addition, OpenACC is used
for explicit memory management to ensure that data for the computations is transferred between CPU and GPU memory as
needed. For a detailed discussion of MPTRAC's MPI-OpenMP-OpenACC parallelization strategy, see Hoffmann et al. (2022).
Notable exceptions to the parallelization strategy of MPTRAC are the random number generation and the particle sorting
module. For random number generation, the GNU Scientific Library (GSL; Galassi and Gough, 2009) is used on the CPUs,
whereas the cuRAND library (NVIDIA, 2023) is used for GPUs. For particle sorting, a key-value sorting algorithm of the
Thrust parallel algorithms library was utilized for the GPUs. For the CPUs, we also considered a key-value sorting algorithm
from the Thrust library, which provides OpenMP parallelization. Since we found that sorting took significantly more time on
the CPUs, we repeated the sweet-spot analysis described in Sect. 6 to find the best tradeoff for the sorting





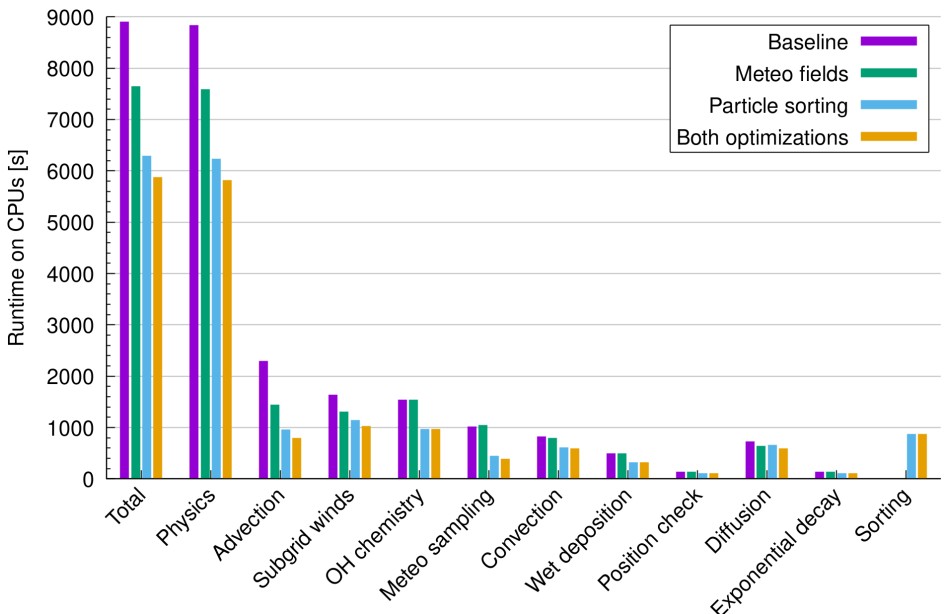

**Figure 9.** Runtime measurements for the largest test case using $10^8$ particles and ERA5 data as input for a CPU-only run on the JUWELS Booster. Similar to Fig. 5, results are shown for the baseline code, for the optimization of the memory layout of the wind and velocity fields, for the sorting of the particle data, and for the full optimization including both methods (see plot key).

time interval $\Delta t_s$ for the CPUs. The analysis showed that for ERA5 data $\Delta t_s$ had to be increased from 1 h for the GPUs to 6 h for the CPUs to achieve the best speedup due to particle sorting.

Figure 9 shows the total runtime, the physics runtime, and the runtime of selected modules for the largest test case of ERA5 transport simulations with $10^8$ particles on the CPUs. This assessment was performed using 12 out of 48 physical CPU
cores, which is the fraction of CPU cores shared by each GPU device on the JUWELS Booster, having four GPU devices per compute node. The CPU-only simulation therefore has the same baseline in terms of CPU cores as the GPU run. The runtime measurements indicate that similar to the OpenACC GPU version, the OpenMP CPU version of the code also benefits significantly from the optimizations discussed earlier. The total runtime is reduced from 8904 to 7650 s ($-14\%$) by optimizing the meteorological data memory layout, to 6290 s ($-29\%$) by particle sorting, and to 5877 s ($-34\%$) by combining both
optimizations. These improvements are largely due to improvements in the physics calculations, which consume more than 99% of the total runtime of the simulations on the CPUs. For the advection kernel, we found the runtime was reduced from 2297 s for the baseline code to 1450 s ($-37\%$) by optimizing the meteorological data memory layout, to 964 s ($-58\%$) by particle sorting, and to 797 s ($-65\%$) by combining both optimizations.

Comparing the runtime of the CPU simulations with the GPU simulations (Figs. 5 and 9) indicates a significantly improved
throughput due to GPU acceleration. Due to the significantly reduced runtime, this would also lead to significant energy savings on the GPUs, despite their higher TDP (Table 1). However, we refrain from further quantifying the GPU-over-CPU speedup




and energy consumption here. Our CPU-only and GPU measurements were performed with a single GPU and with the same baseline in terms of number of CPU cores. However, this does not take into account a potential higher peak performance of the CPUs with respect to better OpenMP scaling and limited OpenACC scaling for multiple GPUs on a compute node. In general, the CPUs of the JUWELS Booster have relatively low computational performance compared to other CPUs of the same technological generation. A CPU-only run to estimate the GPU-over-CPU speedup of MPTRAC should be performed considering an HPC system with more powerful CPUs.

## 8 Summary and conclusions

With unprecedented parallel computing capabilities, GPUs have become indispensable tools for Earth system modeling, enabling faster and more comprehensive simulations that enhance our understanding of complex environmental dynamics. Here, we outline efforts to optimize the MPTRAC Lagrangian particle dispersion model on GPUs, a model that is essential for studying chemical and dynamical processes as well as transport and mixing in the atmosphere. Our optimization efforts primarily targeted MPTRAC's advection kernel, which was found to dominate runtime, in particular for our largest test case comprising $10^8$ particles and using ERA5 meteorological fields to drive the calculations. The focus of our study was on improving simulation runtime and energy efficiency through optimized memory access patterns and data structures, benefiting both GPU and CPU systems.

Detailed performance benchmarks compare baseline and optimized MPTRAC code versions. The test case involved modules to represent air parcel advection, eddy diffusion, subgrid-scale wind fluctuations, convection, exponential decay, and first-order chemical loss processes. Two meteorological reanalyses, ERA5 and ERA-Interim, are used to assess scaling with respect to the size of the input data. Simulations cover 24 h of time with different types of model output and consider problem sizes ranging from $10^5$ to $10^8$ globally distributed air parcels. The benchmarks employ MPTRAC version 2.6, compiled with standard optimization flags for the CPU and GPU versions. The performance benchmarking was conducted on the JUWELS Booster at the Jülich Supercomputing Centre, Germany. The machine features state-of-the-art NVIDIA A100 GPUs and AMD EPYC Rome 7402 CPUs interconnected via third generation NVLINK.

Following initial inspection, the code optimization focused on improving the memory layout and access pattern of the meteorological data and the particle data. The timeline analysis with Nsight Systems identified the advection kernel as the primary bottleneck of the code, requiring 55% of the runtime of the physics calculations of our large ERA5 test case using $10^8$ particles. The roofline and memory analysis of the advection kernel showed that the advection kernel is memory-bound, which further guided our efforts towards data layout and memory access optimizations. The first optimization restructured the layout of the meteorological data, specifically of the horizontal wind and vertical velocity components, for improved cache utilization. Substantial runtime improvements result, with a 44% reduction of runtime of the advection kernel and 48% reduction of the subgrid-scale wind parameterization. Tests showed nearly linear scaling in runtime, with the speedup due to the optimization saturating at a factor of 1.4 as particle count rises.




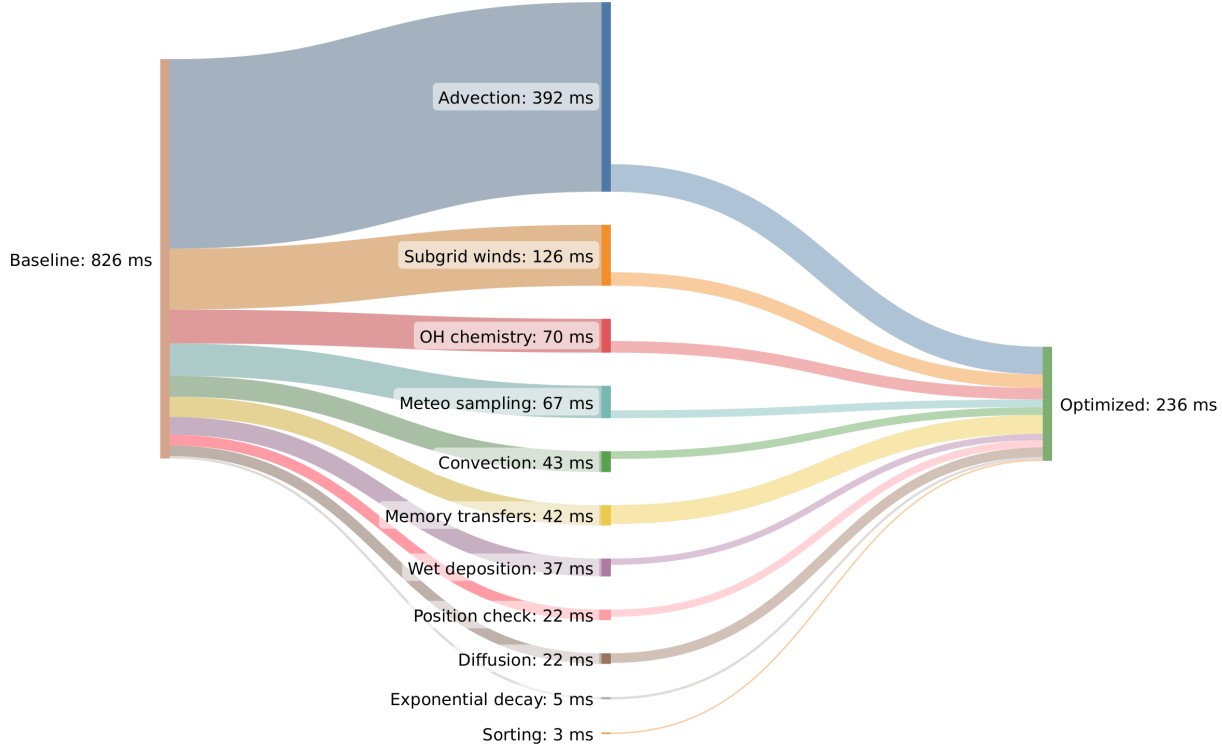

**Figure 10.** Sankey diagram showing the results of our optimization efforts for MPTRAC for the large ERA5 test case with $10^8$ particles. The plot shows the distribution of the mean runtime per time step for the baseline code on the left and for the optimized code on the right. The distribution for the different geophysical modules and components of the code is in the middle. Note that the mean runtime has been normalized in case a module is called several times per time step (e.g., position check of air parcels) or if it is not called at every time step (e.g., meteo sampling, memory transfers, and sorting).

Analyses using Nsight Compute suggested that MPTRAC's advection kernel is affected by near-random memory access
patterns due to near-randomly located air parcels, causing frequent cache misses and performance decline. To address this, a sorting algorithm was introduced as a second optimization, reordering particle data for aligned memory access, reducing cache misses, and therefore enhancing runtime. The sorting process includes computing a linear box index for the particles according to memory alignment, applying a key-value sorting algorithm to find a permutation for reordering, and re-arranging the individual data arrays of the particles according to the permutation. Results demonstrate substantial runtime reductions of
72% for the physics calculations, with a maximum reduction of 80% for the advection kernel. The speedup due to particle sorting grows steadily with increasing problem size as memory caches are used more and more effectively as the number of air parcels per grid box of the meteorological data increases. Combining the memory layout optimization of the meteorological fields with particle data sorting provides further improvements, reducing the runtime of the physics calculations by 75% and of the advection kernel by 85%. Figure 10 summarizes the results of our optimization efforts.



The GPU and CPU codebase of MPTRAC is largely the same, except for the use of OpenMP for CPU parallelization and OpenACC for GPU offloading of the particle loop. However, random number generation requires specific libraries (GSL for CPUs versus cuRAND for GPUs) and different key-value sorting algorithms for CPUs and GPUs, as provided by the Thrust library. The evaluation of the largest ERA5 transport simulations with $10^8$ particles underlines significant runtime improvements of the GPU optimizations also for the CPUs. For our large test case, we found a runtime reduction of 34% for the physics computations, including a reduction of 65% for the advection kernel. The code improvements discussed here are considered to be architecture-independent to a large extent. Although we did not quantify the GPU-over-CPU speedup and energy consumption, as we limited this study to benchmarking the single GPU case, a direct comparison of the runtime measurements generally suggests a significant performance advantage of using GPUs over CPUs.

Optimization of code performance is an ongoing process that does not have a definitive endpoint due to multiple reasons such as complexity of the software, changing requirements, evolving hardware, new algorithms and techniques, and more. For the MPTRAC model, more detailed analyses of the compute kernels and benchmarking on other HPC systems are of interest. Considering the fact that the computations are memory-bound, reducing the size of the meteorological fields by data compression might be promising. Furthermore, kernel fusion might help to increase arithmetic intensity. Nevertheless, the code optimizations discussed here highlight substantial performance benefits with MPTRAC on GPUs and CPUs, bringing the model closer to applications on upcoming Exascale HPC systems.

*Code and data availability.* The MPTRAC model (Hoffmann et al., 2016, 2022) is distributed under the terms and conditions of the GNU General Public License (GPL) version 3. The version 2.6 release of MPTRAC used in this paper is archived on Zenodo (Hoffmann et al., 2023). Newer versions of MPTRAC are available from the repository at https://github.com/slcs-jsc/mptrac (last access: 30 October 2023). The scripts for running the compute jobs and for analyzing and plotting the results of this study are also archived on Zenodo (Hoffmann, 2023). The ERA5 and ERA-Interim reanalysis data products (Dee et al., 2011; Hersbach et al., 2020) were retrieved from ECMWF's Meteorological Archival and Retrieval System (MARS). See https://www.ecmwf.int/en/forecasts/datasets/browse-reanalysis-datasets (last access: 30 October 2023) for more details.

*Author contributions.* Conceptualization: L.H., K.H., A.H., M.H. and J.K.; Formal analysis: L.H., J.C. and M.L.; Investigation: L.H., K.H., J.C. and M.L.; Methodology: L.H., K.H., A.H., M.H. and J.K.; Software: L.H., K.H., J.C. and M.L.; Writing – original draft: L.H.; Writing – review & editing: L.H., K.H., A.H., M.H., J.K., J.C. and M.L.

*Competing interests.* The authors declare that no competing interests are present.



*Acknowledgements.* This study was supported by the Joint Lab Exascale Earth System Modelling (JL-ExaESM) of the Helmholtz Association of German Research Centres. ERA5 data were generated using Copernicus Climate Change Service Information. Neither the European Commission nor ECMWF are responsible for any use that may be made of the Copernicus information or data in this publication. The ERA-Interim data product is released by ECMWF under a Creative Commons Attribution 4.0 International (CC BY 4.0) license. We thank the Jülich Supercomputing Centre for providing computing time and storage resources on the JUWELS supercomputer. We thank Farah Khosrawi, Jülich, for providing helpful feedback and suggestions on an earlier version of this manuscript. We applied DeepL Write for language editing.




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
