# Peer review of "Accelerating Lagrangian transport simulations on graphics processing units: performance optimizations of MPTRAC v2.6"

_EGUsphere, 2023_

## Author Comment (AC1)

**Reply to reviewer comments**

Dear Reviewers, dear Editor,

Thank you for the time and effort spent on the manuscript. We considered the comments and hope that the revised draft properly addresses the open issues. Please find our point-by-point replies below. A revised manuscript with tracked changes has been uploaded.

Kind regards,

Lars Hoffmann

**Reviewer #1**

The manuscript is a follow-up to Hoffmann et al. (2022) where the adaptation of MPTRAC to GPU processing using OPEN-ACC was described and demonstrated. The present work describes two types of optimization producing significant speed-up for both GPU and CPU versions of the code. The manuscript is well written and clear both in the methods and results and should be published.

Thank you for the positive feedback!

I have only a few minor comments and questions to the authors

1) Section 3: Although the scope of this work is technical, a few more words about the type of tracer / molecule and processes considered here would be useful for the sake of completeness.

We agree, and at the beginning of Sect. 3, we added: "The model configuration is similar to previous studies in which we used MPTRAC to simulate the long-range transport and dispersion of sulfur dioxide from volcanic eruptions (Heng et al., 2016; Hoffmann et al., 2016; Wu et al., 2017, 2018; Liu et al., 2020; Cai et al., 2022; Liu et al., 2023)."

2) It is unclear that the ERA5 needs to be used at its maximal spatial and temporal resolution for all transport applications, in particular for large-scale transport. Using the full vertical resolution is certainly a good choice but the horizontal and temporal resolution might be reduced at least for the horizontal wind with limited impact in many cases.

We added more motivation in Sect. 2 as to why we focus on full-resolution ERA5 data in this study: "While downsampling the ERA5 data to a lower horizontal resolution is a viable approach to improve the computational performance of Lagrangian transport simulations, it must also be considered that it introduces transport deviations, for example for convective events in the troposphere (Hoffmann et al., 2019, 2023; Clemens et al., 2023). Therefore, in this study we consider ERA5 data at its full native resolution."

2) l.176: What are NVTX markers? This seems to be a NVIDIA feature for profiling.

We rephrased this and added a reference to clarify:"...the NVIDIA Tools Extension Library (NVTX) ranges (NVIDIA, 2024), which we explicitly added to the code to annotate ranges between two points in the program's execution..."

3) L192: The arithmetic intensity, which is perhaps not a common notion, needs to be defined.

We added the following definition: "The arithmetic intensity is defined as the number of floating point operations divided by the number of bytes of main memory accessed to execute a program."

4) No indication is given about the overlapping of data transfer and calculations. I do not know how this is applicable to the architecture considered here but it is a source of optimization in computers which cache memory. Perhaps it is done automatically but it deserves to be mentioned.

To clarify, we added in Sect. 4: "These data transfers cannot be conducted asynchronously with the computation."

5) I guess that the code is running under a configuration where the nodes are reserved to a single user and not in time-sharing. This also deserves to be mentioned.

Thank you for pointing this out. We added at the end of Sect. 3: "Note that the compute nodes are reserved for a single user and are not time-shared when the test runs are executed. The compute resources of the node are exclusively reserved for the user, while the network and file system access are shared with other users executing jobs on other nodes."

6) Figure 3 is hardly readable. I am not color blind but I see no red and the blue is difficult to distinguish from the green without zooming the figure. It seems that the baseline dots have a green contour and the optimized dots have a blue contour. This figure needs to be improved.

We used an image processing tool to improve the colors of the symbols in this figure.

7) Figure 4: Is it possible to remove the 0.00 B/s channels or to indicate that this is an output from NVIDIA tools that cannot be beautified.

We used an image processing tool to remove the zero entries from the original figure.

8) L 251 : Since two wind fields are required for time interpolation, why not aligning time in a float uvw[EX][EY][EZ][2][3] or float uvw[EX][EY][EZ][3][2] 5-D structure?

This is an interesting idea that could further improve the performance of the code. Unfortunately, it turns out to be difficult to implement in the current version of MPTRAC. During the time loop of the model, we need to update the meteorological data at the synoptic time steps of the reanalyses. Currently, this is accomplished by performing a pointer swap between the data structures for time steps $t_i$ and $t_{i+1}$ in CPU and GPU memory, reading the new data for $t_{i+1}$ into CPU memory, and then transferring it to GPU memory. Adding the time dimension to the meteo data structure would require the data updates

and memory transfers to be implemented explicitly.

9) Is there any impact of aligning the tracer data ?

In the current test case, we would not expect much improvement from aligning the tracer data, because within the individual particle loops of the geophysical modules of MPTRAC, mostly only a single (i.e., particle mass) or a small number of variables are accessed per air parcel. Therefore, we would rather expect a performance degradation when aligning the tracer data. However, this may be different in other applications, e. g., in complex chemistry calculations, where several tracer concentrations per air parcel have to be considered.

10) It is not said whether the sorting is done by copying the tracer arrays or by using a permutation index without moving the data. Certainly the copy has the advantage of avoiding random access to the tracer data for the threads.

As pointed by the reviewer, all data in the arrays must be reordered in memory to avoid quasi-random access. This is a key aspect that needs to be made clear. We therefore extended the description of the sorting algorithm in Sect. 6: "[In the third step, the individual arrays of particle data, i. e., their positions and any additional variables per air parcel, are sorted according to $P$.] Here, the individual data of the arrays are swapped according to $P$, so that the data are aligned and quasi-random access to memory is avoided."

**Reviewer #2**

The manuscript titled "Accelerating Lagrangian transport simulations on graphics processing units: performance optimizations of MPTRAC v2.6" by Hoffmann et al. presents a comprehensive study on the optimization of the MPTRAC Lagrangian particle dispersion model for improved performance on graphics processing units (GPUs). The research focuses on overcoming the challenges associated with the memory-bound nature of the advection kernel in MPTRAC by employing two primary optimization strategies: restructuring the layout of meteorological data and introducing a sorting algorithm for aligned memory access of particle data. The study is methodically sound and well-structured and makes a significant contribution to the field of Earth system modeling, particularly in enhancing the efficiency of Lagrangian transport simulations.

The study not only demonstrates significant advancements in the performance of the MPTRAC model but also provides insights that could be applied to other Lagrangian particle dispersion models. The manuscript is well-written, thoroughly researched, and presents its findings in a clear and accessible manner. It is recommended for publication after considering minor suggestions.

Thank you for the positive feedback!

1) It would be beneficial if the optimization technique regarding the overlapped execution

of data transfers and computations were mentioned.

Data transfers and computations are executed synchronously. Please see reply to Comment 4 of Reviewer #1.

2) A brief discussion on the limitations or challenges faced during the optimization process would add depth to the study.

While some of the particular limitations and challenges of optimizing the code are discussed in Sects. 5 to 8 of the paper, we hope that the reader will find it interesting to read a new paragraph on performance portability, which we see as a major challenge for future work and development of MPTRAC. We added it to the summary and conclusions: "Achieving performance portability is a particular challenge in scientific software development. A performance-portable scientific code should exhibit high levels of performance across different computing platforms, architectures, and hardware configurations, while maintaining its functionality and accuracy. Performance-portable scientific codes allow researchers to run simulations on different HPC systems, ranging from traditional CPUs to specialized accelerators such as GPUs, without sacrificing computational efficiency or accuracy. The current study is limited to some extent in that it focuses on the performance analysis and optimization of MPTRAC on a highly relevant but single HPC system, the JUWELS Booster. We intend to assess and further improve the performance portability of MPTRAC by conducting a more detailed benchmarking study in future work."

3) In Fig. 3, the "red" dots are difficult to distinguish from the "orange" dots. The figure would be more readable if there were more obvious color differences between "red" and "orange" and between "blue" and "green".

We used an image processing tool to improve the colors of the symbols in this figure.

4) In Fig. 4, some words, for instance, "* Global *" and "* Compression," are not fully visible. I guess the figure might be produced by the NVIDIA software, but it would be beneficial to address this display issue for clarity, though this does not affect the integrity of the results presented.

Unfortunately, we were unable to fix this problem with the text labels in the figure. The figure was created directly with the Nsight Compute tool, which has limited capabilities for changing font size. However, as the reviewer suspected, the parts of the figure affected by this problem are not really relevant to the results or discussion of this study.

5) As a reader, I am interested in the specific method of sorting the data, whether they are sorted by assigning an index to the arrows or creating a new set of data based on the target elements.

We extended the description of the sorting method in order to clarify that the individual data of the arrays are rearranged (swapped) in memory in order to achieve better alignment and avoid quasi-random access patterns. Please see reply to Comment 10 of Reviewer #1.

**Reviewer #3**

The manuscript provides detailed memory access optimization schemes for the Lagrangian particle dispersion model MPTRAC (Hoffman et al., 2022) especially on GPUs. The motivations of the two optimization schemes are clearly analyzed and the optimization processes are thoroughly demonstrated and tested, making the research sound and rigorous. The use of the Array of Structures method and the particle data sorting method provides new insights into memory optimization for earth system model simulations on GPUs.

The manuscript is well written and should be published after consideration for some minor questions.

Thank you for the positive feedback!

1) The color of the dots in Fig. 3 is hard to distinguish. It's best if more explanations of the figure can be provided about the result comparison of two model versions.

We used an image processing tool to improve the colors of the symbols in this figure.

2) How the performance of the optimization scales with problem size has been investigated in the research. I'm curious whether the tests were also run on other number of GPU cores and whether the optimized model shows similar time improvement.

On the GPUs, one cannot easily restrict the number of cores, but one can set the number of blocks and threads per block. The product of these can be set to match the number of cores to be used. However, it is not possible to control where the blocks are scheduled, hence they may or may not be packed on the same stream multiprocessor (SM) units. This leads to idle cores on some SMs if the blocks are spread out across the SMs. In general, for performance, it is best to not restrict the total number of blocks and instead base this on the problem size.

In this study, we focused on performance analysis and optimization of MPTRAC on the NVIDA A100 GPUs of the JUWELS Booster as it is currently the most relevant HPC architecture for us. In the introduction, we added: "Here, we focus on performance analysis and optimization on the JUWELS Booster as this is the state-of-the-art production machine at the Jülich Supercomputing Center, Germany, on which most of the simulations with MPTRAC are currently being run."

Only recently, we tested the optimizations on the NVIDIA Grace Hopper 200 architecture, which also showed significant performance improvements due to the code optimizations discussed in the paper. Porting and performance analysis on GPUs from other vendors is also in progress. While this is preliminary work and not reported in the paper, it gives us confidence that the code optimizations discussed here will be relevant to other HPC systems as well.

3) Data communications among GPU cores are commonly necessary in parallel sorting. I'm interested how much communication is introduced in the particle sorting through the

Thrust library and how does is vary with the number of GPUs used. Does it become notable under certain GPU number settings?

We are afraid that we cannot provide the specific details of the implementation of the Thrust library's sorting algorithm that the reviewer requests here, other than the reference provided in the paper. As discussed in our response to Comment 2, we refrain from explicitly changing the number of blocks and threads per block on the GPU devices, but use them in the configuration provided by the Thrust implementation.

4) I assume that the simulation results of the optimized model are bitwise identical with the base model, which is commonly achieved in model optimization works. I think it's worth mentioned in the manuscript.

This is an important aspect, thank you for pointing it out. For the optimization of the meteo data memory layout, we found bitwise identical results. In Sect. 5, we added: "Note that the simulation results from the optimized and original version of the code give bitwise identical results, as expected." However, for the sorting method the results are not bit-wise identical. In Sect. 6, we added: "Note that the simulation results from the particle sorting code and the base version are merely statistically the same. This is due to the fact that the reordering of the particle data means that individual particles are assigned different random perturbations in the diffusion and convection modules during the course of the simulations, because the stream of random numbers was not reordered accordingly."

**References**

Cai, Z., Griessbach, S., and Hoffmann, L.: Improved estimation of volcanic $SO_2$ injections from satellite retrievals and Lagrangian transport simulations: the 2019 Raikoke eruption, Atmos. Chem. Phys., 22, 6787–6809, doi: 10.5194/acp-22-6787-2022, 2022.

Clemens, J., Hoffmann, L., Vogel, B., Grießbach, S., and Thomas, N.: Implementation and evaluation of diabatic advection in the Lagrangian transport model MPTRAC 2.6, Geosci. Model Dev. Discuss., 2023, 1–38, doi: 10.5194/gmd-2023-214, 2023.

Heng, Y., Hoffmann, L., Griessbach, S., Rößler, T., and Stein, O.: Inverse transport modeling of volcanic sulfur dioxide emissions using large-scale simulations, Geosci. Model Dev., 9, 1627–1645, doi: 10.5194/gmd-9-1627-2016, 2016.

Hoffmann, L., Rößler, T., Griessbach, S., Heng, Y., and Stein, O.: Lagrangian transport simulations of volcanic sulfur dioxide emissions: Impact of meteorological data products, J. Geophys. Res., 121, 4651–4673, doi: 10.1002/2015JD023749, 2016.

Hoffmann, L., Günther, G., Li, D., Stein, O., Wu, X., Griessbach, S., Heng, Y., Konopka, P., Müller, R., Vogel, B., and Wright, J. S.: From ERA-Interim to ERA5: the considerable impact of ECMWF's next-generation reanalysis on Lagrangian transport simulations, Atmos. Chem. Phys., 19, 3097–3124, doi: 10.5194/acp-19-3097-2019, 2019.

Hoffmann, L., Konopka, P., Clemens, J., and Vogel, B.: Lagrangian transport simulations using the extreme convection parameterization: an assessment for the ECMWF reanalyses, Atmos. Chem. Phys., 23, 7589–7609, doi: 10.5194/acp-23-7589-2023, 2023.

Liu, M., Huang, Y., Hoffmann, L., Huang, C., Chen, P., and Heng, Y.: High-Resolution Source Estimation of Volcanic Sulfur Dioxide Emissions Using Large-Scale Transport Simulations, in: International Conference on Computational Science, pp. 60–73, Springer, doi: 10.1007/978-3-030-50420-5$_5$, 2020.

Liu, M., Hoffmann, L., Griessbach, S., Cai, Z., Heng, Y., and Wu, X.: Improved representation of volcanic sulfur dioxide depletion in Lagrangian transport simulations: a case study with MPTRAC v2.4, EGUsphere, 2023, 1–29, doi: 10.5194/egusphere-2022-1480, 2023.

NVIDIA: NVTX (NVIDIA Tools Extension Library), URL `https://github.com/NVIDIA/NVTX`, last access: 2 April 2024, 2024.

Wu, X., Griessbach, S., and Hoffmann, L.: Equatorward dispersion of a high-latitude volcanic plume and its relation to the Asian summer monsoon: a case study of the Sarychev eruption in 2009, Atmos. Chem. Phys., 17, 13 439–13 455, doi: 10.5194/acp-17-13439-2017, 2017.

Wu, X., Griessbach, S., and Hoffmann, L.: Long-range transport of volcanic aerosol from the 2010 Merapi tropical eruption to Antarctica, Atmos. Chem. Phys., 18, 15 859–15 877, doi: 10.5194/acp-18-15859-2018, 2018.